# Smart epidemic control: A hybrid model blending ODEs and agent-based simulations for optimal, real-world intervention planning

**Péter Polcz**[1]*, **István Z. Reguly**[1,2], **Kálmán Tornai**[1], **János Juhász**[1,3], **Sándor Pongor**[1], **Attila Csikász-Nagy**[1,2], **Gábor Szederkényi**[1]

**1** National Laboratory for Health Security, Faculty of Information Technology and Bionics, Pázmány Péter Catholic University, Budapest, Hungary, **2** Cytocast Hungary Kft., Budapest, Hungary, **3** Institute of Medical Microbiology, Semmelweis University, Budapest, Hungary

* polcz.peter@itk.ppke.hu

**Data availability statement:** Package PanSim and the additional Matlab code are available in

## Abstract

Optimal intervention planning is a critical part of epidemiological control, which is difficult to attain in real life situations. Ordinary differential equation (ODE) models can be used to optimize control but the results can not be easily translated to interventions in highly complex real life environments. Agent-based methods on the other hand allow detailed modeling of the environment but optimization is precluded by the large number of parameters. Our goal was to combine the advantages of both approaches, i.e., to allow control optimization in complex environments. The epidemic control objectives are expressed as a time-dependent reference for the number of infected people. To track this reference, a model predictive controller (MPC) is designed with a compartmental ODE prediction model to compute the optimal level of stringency of interventions, which are later translated to specific actions such as mobility restriction, quarantine policy, masking rules, school closure. The effects of interventions on the transmission rate of the pathogen, and hence their stringency, are computed using PanSim, an agent-based epidemic simulator that contains a detailed model of the environment. The realism and practical applicability of the method is demonstrated by the wide range of discrete level measures that can be taken into account. Moreover, the change between measures applied during consecutive planning intervals is also minimized. We found that such a combined intervention planning strategy is able to efficiently control a COVID-19-like epidemic process, in terms of incidence, virulence, and infectiousness with surprisingly sparse (e.g. 21 day) intervention regimes. At the same time, the approach proved to be robust even in scenarios with significant model uncertainties, such as unknown transmission rate, uncertain time and probability constants. The high performance of the computation allows a large number of test cases to be run. The proposed computational framework can be reused for epidemic management of unexpected pandemic events and can be customized to the needs of any country.

the public repository
https://github.com/khbence/pansim. Otherwise,
this research includes solely synthetic data,
which can be produced using PanSim and the
supplementary Matlab scripts executing
PanSim.

**Funding:** The study was funded by the National
Research, Development, and Innovation (NRDI)
Office in Hungary (RRF-2.3.1-21-2022-00006)
and the Ministry of Culture and Innovation of
Hungary from the NRDI Fund through the
grants PD-145902 (P.P.), K-145934 (G.Sz.),
and FK-145931 (I.Z.R.) financed under the
OTKA Research Thematic Call 2023. We
acknowledge the support of Ministry of Culture
and Innovation of Hungary through the New
National Excellence Program scholarship
ÚNKP-23-4-II-PPKE-27 (P.P.). Moreover,
projects no. TKP2021-NKTA-66 (P.P.) and
TKP2021-NVA-26 (P.P.) have been
implemented with the support provided by the
Ministry of Culture and Innovation of Hungary
from the NRDI Fund, financed under the
TKP2021-NKTA and TKP2021-NVA funding
schemes, respectively. This work was also
supported by the Hungarian Academy of
Sciences through the Grant
POST-COVID2021-64 (I.Z.R.). The funders had
no role in study design, data collection and
analysis, decision to publish, or preparation of
the manuscript.

**Competing interests:** We have read the
journal's policy and the authors of this
manuscript have the following competing
interests: BK-H and AC-N reporting that they
have financial and business interests in
Cytocast Kft. that may be affected by the
research reported in the enclosed paper.

## Author summary

Pandemic control is a complex task that requires balancing the impact of interventions like lockdowns or closures of institutions with the need to limit the spread of disease. In this study, we developed a new tool that combines two powerful approaches: mathematical modeling to predict the spread of the disease and detailed simulations that reflect real-world conditions. By combining these methods, we were able to create a strategy that efficiently manages an epidemic similar to COVID-19. Our approach was not only effective in controlling the outbreak but also worked well even when there were uncertainties about how fast the virus spreads. We believe that the proposed method of combining agent-based and ODE models, will inspire research communities working in other scientific fields involving agents, Cell Biology, Transportation, or Logistics.

## Introduction

Designing appropriate measures to meet often conflicting goals and constraints during the COVID-19 pandemic was challenging for most countries, especially in 2020 [1]. Moreover, we have seen that any delay in intervention could cause serious damage [2–4]. Beyond varying degrees of success in containing the epidemic, there was a palpable sense of panic among the population, which was even worsened by a lack of accurate information about the local situation. This emergency demonstrated that local and national governments need to be equipped with a comprehensive pandemic management system with monitoring, prediction, and decision support capabilities to keep epidemic spread under control at multiple levels of the community, e.g., in settlements [5], but also at a national and international level [6]. Such an epidemic surveillance and modeling framework has to be able to cope with uncertainties coming from the lack of data, unmodeled phenomena, or the appearance of yet unknown pathogen variants.

Over the years, analysis, estimation, and planning based on dynamic models have become essential parts of epidemic management [7]. Moreover, the recent pandemic provided unprecedented momentum to computational epidemiology worldwide (see, e.g., the review papers [8,9] and references therein). Notably, important dynamic analysis results are available for epidemic processes [10–12]. Furthermore, numerous epidemic surveillance tools were developed to solve particular problems related to the COVID-19 pandemic, such as state and parameter estimation [13,14], or the retrospective assessment of certain *non-pharmaceutical interventions* (NPI) [15]. There is also an extensive body of literature on predicting the future evolution of disease spread, as demonstrated in the review paper [16]. Although several machine learning approaches are available for prediction [17] and intervention planning [18], compartmental *ordinary difference equation* (ODE) models are still widely used and remain highly popular for managing pandemics at the population level [8]. These models make it possible to design robust and/or optimal control strategies to mitigate the impact of the pandemic (see, e.g., [19–21]). In addition, *model predictive control* (MPC) strategies allow the formulation of complex control objectives [22–24] through the minimization of a cost function, which can reflect many aspects of the future evolution of a spread, e.g., meeting hospital capacity, minimizing the stringency of interventions, etc.

The majority of the model-based control strategies are designed for SEIR-type compartmental models. A SEIR model describes the dynamics of *susceptible*, *exposed*, *infected*, and *recovered* people. Basic SEIR models can be enhanced in several ways, e.g., they can be augmented with additional phases or states of the disease, and compartments can be

split by age groups, vaccination status or by locations. Although compartmental models can be extended to include a high level of detail, they still present several issues and limitations, two of which are critical to our work. First, model-based controllers provide control actions such as lockdown [18] or quarantine measures [23,25]. These are often combined with screening measures [21,24], or vaccination intensity [20]. Since these measures take values in continuous ranges, it can be difficult to translate them into realistic interventions. Secondly, low-dimensional compartmental models have limited prediction accuracy in heterogeneous communities [26]. On the other hand, detailed models require a large number of parameters and state variables, which are often difficult to determine from available measurements.

Another large group of results in pandemic management is related to *agent-based models* (ABMs) [27–29], which enable a fine-grained study of disease spread. Although statistical [30], compartmental [31,32], and other model-based [33] techniques have been used to assess the impact of interventions, ABMs have proven particularly effective in capturing the nuanced impact of NPIs [34–36].

Some ABMs, such as a soil-crop irrigation model in [37], can be expressed in closed-form dynamic equations, making them amenable to advanced control techniques, including model predictive control (MPC). However, agent-based epidemic models that simulate real-world NPIs [38] are significantly more complex, making direct integration with model-based control algorithms challenging. As a result, most agent-based decision support systems lack fully automated intervention planning.

To deal with the complexity of agent-based simulators, surrogate models are often used to approximate their behavior, making statistical inference [39], calibration [40], or sensitivity analysis [41] easier and more efficient. One of the early works to recognize the potential of surrogates in predictive control design for agent-based models (ABMs) is presented by An et al. [42]. It proposed a novel framework for the predictive control of agent-based simulators. By approximating the simulators behavior with mathematically more tractable models, the study laid the groundwork for the multi-level optimization-based decision-making in complex real-world processes with agents. This concept has been further developed by Niemann et al. [43,44], who proposed gradient-based optimization methods for the multi-level predictive control for agent-based epidemic models.

For agent-based epidemic spread simulators, it is natural to choose a SEIR-type ODE surrogate, however, fitting a model to a general ABM is often challenging. A data-driven approach is proposed in [43], where Koopman operators are trained using simulation data to fit linear models to ABM. Instead, Fonseca et al. [45] have considered a few classes of nonlinear system (e.g., S-systems [46]), and investigate, how they can be fit to the targeted ABM.

Despite the rapid development and spotlight on computational epidemiology, there are existing open problems to solve. In addition to the challenges already mentioned, Alamo et al. [28] highlighted the difficult multi-criteria nature of real and practically implementable intervention planning. The available ABMs and new results in MPC design, as well as the gaps in the state of the art, motivated this research on the intelligent combination of different modeling approaches to support complex decision-making.

As the key contribution of the paper, we propose a computationally efficient framework that computes real-word implementable NPIs in terms of six measures, mask-wearing rules, quarantine policies, testing intensity, curfew, school closure, or closing public venues. The stringency of these measures belong to a discrete set of values ("low", "medium", "high"), which correspond to the actually implemented interventions in Hungary or other countries during the COVID-19 pandemic between 2020 and 2022. Computing optimal combinations of these measures allow us to achieve a wide variety of epidemic prevention or mitigation

targets, even following a targeted epidemic reference curve. To achieve these results, we solve an important research gap in epidemic management by constructing a statistical model from synthetic data to map the targeted transmission rate to a set of combinations of measures. Synthetic data was generated using PanSim, a GPU-accelerated, open-source microsimulation model developed by our research group [5]. PanSim provides a realistic, age-stratified, georeferenced simulation of 180,000 agents across 81,000 locations, operating with a relatively small, a 10-minute time step. The statistical model, combined with the ODE-based robust MPC framework and the fine-grained yet accelerated PanSim, enables an effective integration of model-based control with a highly detailed agent-based simulation. This fusion leverages the strengths of both approaches: a mathematically rigorous control strategy capable of optimizing diverse cost functions and a high-resolution model that accurately represents spatial and temporal population dynamics, facilitating the direct implementation of NPIs.

The paper is organized in three main sections. First, we present the main findings in section Results, which are illustrated through seven case studies labeled by capital letters **A**–**G**. In section Discussion, we summarize the advantages and limitations of our approach and suggest possible research directions for the future work. We present the MPC methodology in detail in section Methods.

## Results

For pandemic control through non-pharmaceutical intervention planning, we consider two distinct models, an agent-based and a compartmental ODE model, each characterized by unique properties and limitations. The ABM boasts high fidelity, rendering it capable of accurately simulating the effects of specific NPIs. However, its drawback lies in its non-closed form nature, posing challenges in planning control inputs based on this model. Conversely, an ODE model is endowed with a closed-form structure, facilitating model-based control computations. Nevertheless, it lacks the ability to simulate the effects of specific interventions. To harness the strengths of both models, an interface between the two models can be devised to translate control inputs intended for the ODE model into practically implementable control strategies for ABM.

We make use of a high fidelity fine-grained microsimulation model, to which we propose MPC strategies. The framework computes a targeted transmission rate ($\beta$) for the pathogen, then, it translates to an appropriate combination of NPIs, which is simulated on the ABM. Possibly, the simulator shows a significantly different transmission rate than the targeted one. In that case, the mapping from the transmission rates to NPIs ($\beta \mapsto$ NPI) can be updated optionally.

We examine the combinations of six predefined *policy measures*, which are described in Table 1. A combination of the six policy measures is considered a non-pharmaceutical intervention (NPI). For instance, the combination (TP-medium, PL-high, CF-low, SC-high, QU-medium, MA-high) is one of the possible $3 \cdot 2 \cdot 2 \cdot 3 \cdot 3 \cdot 2 = 216$ NPIs. During the intervention planning, we allowed any possible combinations of these policy measures, having different effects on the transmission rate. Possibly not all the 216 NPIs are reasonable or practically useful, but those NPIs can be sorted out at any stage of the workflow.

Note that the effects of the six interventions in a combination are generally neither additive nor multiplicative, hence, it is difficult to quantify the overall effect of an NPI on the transmission rate. For this reason, a lookup-table will be introduced later to model the effects of NPIs.

**Table 1. Description of policy measures considered in PanSim.**

| Label | Description |
|---|---|
| TP | Testing intensity: 0.5% (low), 1.5% (medium), 3.5% (high) of population per day. |
| PL | Public venues to close (high) or not (low). |
| CF | Curfew between 8 pm and 5 am enabled (high) or not (low). |
| SO | School closures: up to 3rd grade (medium), up to 12th grade (high), or no closure (low). |
| QU | Quarantine policy: only detected infected individuals are quarantined (low), detected individuals and household members are quarantined (medium), detected individuals, household members, and workplace/school contacts are quarantined (high). |
| MA | Mask wearing rules: the transmission rate is decreases by 20% in enclosed areas when mask are mandatory there (high), otherwise, by 0%, when masks are not mandatory. |

## PanSim, a detailed agent-based simulator

The development of PanSim [5] started shortly after the creation of the Hungarian COVID-19 Mathematical Modelling and Epidemiological Analysis Task Force at the beginning of the pandemic in March 2020. The goal was to complement ODE-based models and generate fine resolution high quality synthetic data [47] by capturing finer details of the spread of the epidemic in realistic, age-stratified populations, and thereby allow the direct evaluation of the effects of non-pharmaceutical interventions. PanSim allows for the direct, realistic implementations of the previously defined interventions because it directly simulates the movement of agents to various locations and allows adding rules to modify these behaviors. Therefore, the effect of interventions on the spread of the infection (the reproduction rate) can be simply calculated from the simulation output; this is in contrast to compartmental models, where the change in the reproduction rate due to a given intervention is captured in a time-varying parameter which is difficult to quantify.

PanSim was set up to simulate a mid-sized Hungarian town (Szeged) using realistic statistics on the population, the points of interest the agents can visit (schools, offices, shops, hospitals, and more), as well as their daily movements. This setup includes 180k agents and 81k locations. The simulation has a 10-minute time step (i.e., state of each agent is updated in every 10 minutes) and probabilistically infects agents at their current location, depending on the number of infectious agents there. Agents follow a probabilistic daily schedule, which is potentially altered by interventions/restrictions currently active - these can be updated daily. The simulator calculates various statistics at midnight of each simulated day, including the number of agents in different stages of the disease by age group and the number of tests, their positivity, the number of agents in quarantine, daily vaccinations, and many more.

PanSim was initially implemented and parameterized for the wild-type SARS-CoV-2. Later, it was easily adapted to the Alpha, Delta, and Omicron variants by updating disease-specific parameters stored in JSON and spreadsheet files. The model can also be extended to other respiratory pathogens, such as influenza virus, respiratory syncytial virus (RSV), or measles virus. PanSim was successfully used extensively during 2021 and 2022 to forecast the spread of different variants and to evaluate the efficacy of various interventions. The hospital load was calculated with less than 10% error 3–4 weeks in advance, which greatly helped intervention planning. Although hospitalizations typically lag infections, this delay does not necessarily degrade the quality of the prediction. The time evolution from infection to hospitalization is inherently captured through the time constants governing compartmental transitions, which will be introduced in the next section. While alternative indicators, such as test positivity rates [24], could also provide valuable predictive insights, hospital data remained

the most reliable source throughout the first two years of the COVID-19 pandemic in Hungary. From 2022 onward, the integration of nationwide wastewater analysis further improved the accuracy and robustness of epidemic predictions [48].

## Compartments of the epidemic models

We reuse the compartmental description proposed by [23], wherein the population of **N** individuals is partitioned into eight distinct groups. The susceptible population (**S**) encompasses individuals who have never been infected. The recovered individuals (**R**) consist of those who have acquired immunity through recovery. The infected population is further subdivided based on various phases of the disease and the potential outcomes.

Specifically, individuals are categorized into the latent phase (**L**), presymptomatic phase (**P**), and the main sequence of the disease. It is noteworthy that the incubation period corresponds to the sum of duration of the latent and presymptomatic phases. The distinction between **L** and **P** is justified by the fact that individuals in the latent phase are not yet infectious, whereas those in the presymptomatic phase may transmit the infection.

To depict the potential disease outcomes during the main sequence of the disease, we delineate four severity scenarios, each parameterized by three probability coefficients. Firstly, a subset of individuals remains asymptomatic (**A**), while others exhibit symptoms (**I**) with a probability $p_I$. Individuals displaying symptoms may require hospitalization (**H**) with a probability $p_H$, and among hospitalized patients, there is a possibility of mortality (**D**) with a probability $p_D$.

The average residence time in compartments **L**, **P**, **I**, **A**, **H** is denoted by the reciprocals of the time constants $\tau_L$, $\tau_P$, $\tau_I$, $\tau_A$, $\tau_H$, respectively.

To make the agent-based model suitable for clinical purposes, severe cases are treated separately by the simulator. Therefore, the compartments **I**, **H**, **R** in the simulator are each split into two disjoint compartments $(\mathbf{I}_m, \mathbf{I}_s)$, $(\mathbf{H}_m, \mathbf{H}_s)$, and $(\mathbf{R}_r, \mathbf{R}_h)$, respectively, where $\mathbf{I}_m$ and $\mathbf{H}_m$ denotes the infected and hospitalized people with mild symptoms, whereas, $\mathbf{I}_s$ and $\mathbf{H}_s$ denote those who show severe symptoms. Severe cases remain in hospital $(\mathbf{R}_h)$ for complications or observation even after recovery, others recover and are sent home $(\mathbf{R}_r)$.

The transitions between the eight compartments of the ODE model are illustrated in purple box of Fig 1, whereas, the gray box illustrates the transitions between the eleven compartments of the ABM. Fig 1 also shows that the compartmental model behind the ABM is a more detailed version of the ODE.

To synchronize the states of the agent-based and the ODE models, we use $\mathbf{I}_k$ as a primary measure. This fact is illustrated in Fig 1 by the red link between compartments $(\mathbf{I}_m, \mathbf{I}_s)$ the ABM (purple box) and **I** of the ODE model (gray box)

In a real pandemic situation, the accurate number of infected people with symptoms in the main phase of the disease $(\mathbf{I}_k)$ is typically unknown. However, previous work demonstrated that the number of active cases can be inferred from the hospital load and the admission rate [49–51] possibly in combination with wastewater data [48]. Using $\mathbf{H}_k$ alone to match the two models may compromise accuracy as $\mathbf{H}_k$ have longer and more distributed delays and lower values or fluctuations. We have decided, therefore, to link the two models through $\mathbf{I}_k$, which is sufficiently high and representative during the course of a pandemic wave, such that the stochastic nature of the simulation (and a real epidemic event) does not significantly affect $\mathbf{I}_k$ even for a smaller population like Szeged.

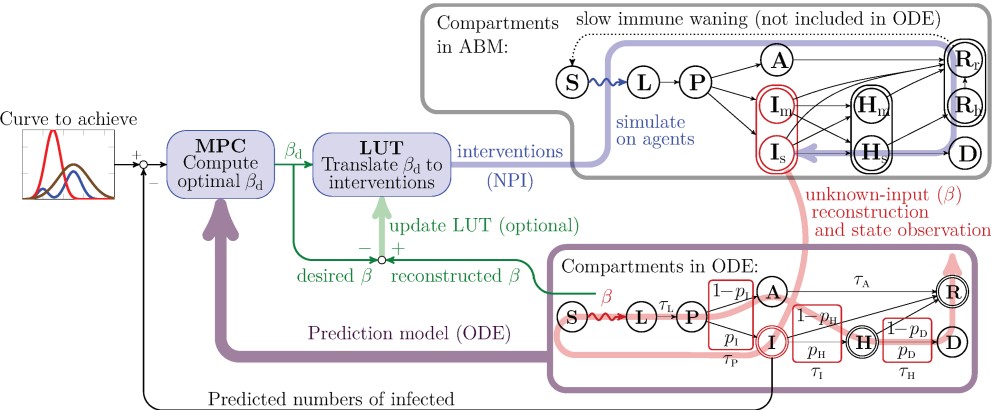

**Fig 1. Block diagram of the proposed epidemic control loop, including the flow diagrams between compartments in the agent-based (gray box) and the ODE models (purple box).** The thick purple arrow illustrates that the ODE model is used as a prediction model in the MPC. The blue arrow shows that selected interventions are simulated on the agent-based model (or applied to a real pandemic event) over a period of $h = 7, 14, 21,$ or 30 days, called a *cycle*. The red arrow illustrates the fact that the state and the transmission rate of the simulator (or a real epidemic) is estimated using the available data. The green arrows indicate that the lookup table (LUT) is updated whenever the reconstructed transmission rate is significantly higher than as it was expected for the selected NPI.

## Assumptions

In this section, we introduce a few considerations to simplify epidemic modeling. First, as reported by [52,53], individuals who are asymptomatic during the main phase of infection exhibit lower infectiousness compared to those in the presymptomatic phase (**P**) or individuals symptomatic during the main phase (**I**). Additionally, we do not exclude the possibility of transmitting the disease in hospitals. Accordingly, we assume that only a fraction $q_A$ of asymptomatic individuals and $q_H$ of hospitalized patients are infectious. Therefore, the number of infectious people at time $k$ is

$$z_k = \mathbf{P}_k + \mathbf{I}_k + q_A \mathbf{A}_k + q_H \mathbf{H}_k. \tag{1}$$

Secondly, we assume that the effects of the non-pharmaceutical interventions are time-invariant. Namely, a NPI will result in approximately the same transmission rate drop of the pathogen at any phase of an outbreak (e.g., rise, peak, or fall) independently of the virus variant. This assumption is reasonable as the social distancing measures reduce the number of contact events by a given scaling factor regardless of virus mutations or epidemic phase. We also note that time-invariance for PanSim was demonstrated in [54] when the spread is caused by a single virus strain.

Thirdly, we assume that a realistic simulator is already set up and ready to use. Namely, this paper does not deal with the model calibration of an agent-based model, as the parameter calibration of PanSim was already addressed in [5,55].

Finally, we assume that only patients in hospitals with severe symptoms are at risk of dying from the disease. This assumption implies, naturally, that severely ill patients are admitted to hospital. We do not model the natural deaths and births in this relatively short period.

It is reasonable to assume that the average residence times $\tau_L^{-1}, \tau_P^{-1}, \tau_I^{-1}, \tau_A^{-1}$ in compartments **L**, **P**, **I**, **A**, the probability of symptomatic infection ($p_I$), and the relative transmission rate of asymptomatic individuals ($q_A$) depend primarily on the properties (e.g., virulence and

infectiousness) of the pathogen, making them approximately constant during the dominance of a single variant. On the other hand, parameters such as the average length of hospitalization ($\tau_H$), the relative infectiousness of hospitalized patients ($q_H$), the probability of hospitalization ($p_H$), and the probability of fatal outcomes ($p_D$) could be influenced by external factors like hospitalization policies or masking rules in hospitals. However, in this study, these factors are not varied. For instance, masking rules (MA) in both *low* and *high* policy levels always assume mask-wearing in hospitals. Thus, we consider $\tau_H$, $p_H$, $p_D$, and $q_H$ to remain constant during the dominance of a single virus variant.

### Preparing steps of the workflow

Although, the workflow is described in detail in Section *Methods*, we think useful to present the major building blocks of our control strategy in brief.

The ODE-based epidemic controllers (see, e.g., [23]) usually compute an action in terms of a single scalar value, the desired transmission rate $\beta_d$, that should be implemented through interventions. The question arises, which interventions are to be selected to achieve the target rate $\beta_d$? Obviously, the simulator was specifically designed to quantify the effects of NPIs even in terms of the transmission rate $\beta$, therefore, the mapping NPI $\mapsto \beta$ is simple to approximate from the simulation data. However, the inverse mapping is more challenging to describe.

In this work, we demonstrate that even a simple statistical model in the form of a lookup table (LUT) is sufficient to capture the mapping from $\beta$ to NPI. To construct the LUT, we generate synthetic data by simulating the effects of every possible NPIs in different phases of the disease as proposed in [55]. Then, the LUT is filled in with the averages and the standard deviations of the resulting transmission rates. In this way, a small amount of data and expert knowledge about a specific pathogen are sufficient for the fine-grained simulator to generate useful synthetic data, considering various possible scenarios that have not been tested in real life. It is also worth mentioning that the use of synthetic data in epidemiology was already discussed in the literature, and it was portrayed as a promising approach [47].

### Control loop

When designing the controller, we must take into account that it is not feasible to introduce new measures on a daily basis. Therefore, the controller is forced to plan a sequence of transmission rates, which does not change within an *intervention planning cycle* having a length of $h$ days. In this way, the targeted transmission rate function is a piecewise constant function, which can be directly integrated into a model predictive controller synthesis. An intervention planning cycle is a period of $h = 7, 14, 21$, or $30$ days, during which the measures to suppress the spread cannot be changed.

We consider a target epidemic curve, which we want to achieve using the interventions. The target curve is an important element of the control design as it allows to shape the time evolution of the spread. Through the target curve we can prescribe, e.g., how much we want to flatten the curve of the spread compared to the free spread.

In the controller loop (Fig 1), first, we presume an initial state, in which the majority of the population is susceptible and only a few people are infected. A model predictive controller with an ODE prediction model computes a sequence of transmission rates, one value for each future intervention planning cycle. Using the LUT, we select an NPI, which is expected to result in a rate close to the desired $\beta_d$. We remark that a combination of measures achieving a given effect is not unique. However, this flexibility is not a limitation; rather, it allows for selecting combinations based on heuristics or specific objectives. For instance, criteria can be introduced to minimize changes in interventions across the intervening cycles.

Then, we simulate the selected NPI for the next cycle (i.e., for $h = 7, 14, 21,$ or 30 days) and collect the simulation's data. By the end of the cycle, we reconstruct the evolution of the spread in the simulator to match the ODE model. The reconstructed data includes the actual state and the rate $\beta$ that we actually achieved with the selected NPI. Closing the loop, the new state is fed back and considered during the next MPC execution.

As an optional step by the end of an intervention planning cycle, we are allowed to update the LUT if it seem reasonable. E.g., when the ongoing epidemic is not well-characterized by the simulator, the transmission rate values in the LUT may not be precise enough. It may also happen that a new variant emerges with a higher transmission rate. In these situations, we provide the opportunity to scale up the transmission rate values in the LUT. Later, we will illustrate the effectiveness of this tuning knob.

We will use the simulator to demonstrate the adaptivity of our approach in possible unexpected scenarios. However, if the situation requires so, the proposed control loop can also be applied to a real epidemic situation. In that case, the unknown sequence for **I** can be computed by our wastewater-based reconstruction method presented in [48], which was successfully applied during the outbreaks caused by the Delta and the Omicron variants in 2022 and 2023.

## Case studies of expected scenarios

In this and the following sections, we analyze multiple scenarios and test the controller algorithm with different pandemic management strategies.

We considered multiple epidemic control goals to evaluate the controller when the pathogen does not go under mutations. In these cases, the controller is well prepared for the emergent pathogen, its transmission rate is precisely estimated and the disease course dynamics is well modeled. We considered a pathogen, which resembles the original SARS-CoV-2 virus strain in both transmissibility and virulence.

When the virus is free to spread without interventions, the epidemic peaks at 1950 in a population of 100k individuals. The obtained curves for the free spread and their deviation are shown in Fig 2. Considering the epidemic curve of a free spread, which is illustrated by green curves in Fig 2, we defined four possible strategies to mitigate the impact of the outbreak:

  **A.** flatten the curve to avoid hospital overload
    (**A1**–**A4** in Fig 2),
  **B.** delay the outbreak to increase hospital capacity (**B1**–**B4**),
  **C.** minimize economic impact (**C1**–**C4**),
  **D.** allow higher mobility for special occasions (**D1**–**D4**).

We want to achieve these control objectives with an intervention planning cycle of length at least $h = 21$ days.

In all case studies (**A**–**D**), we executed PanSim 20 times both when the virus was allowed to spread freely and when the outbreak was controlled by MPC. In Fig 2, all the 20 curves are presented for all cases in gray. The 95% confidence intervals for all curves are illustrated by the colored shaded areas: orange for controlled spread, green for the free spread, and blue for the partially free spread.

Although in PanSim 180k agents are considered, the computed numbers are all normalized to a 100k population. In all figures, we illustrate the normalized curves, i.e., the number of individuals in a 100k population.

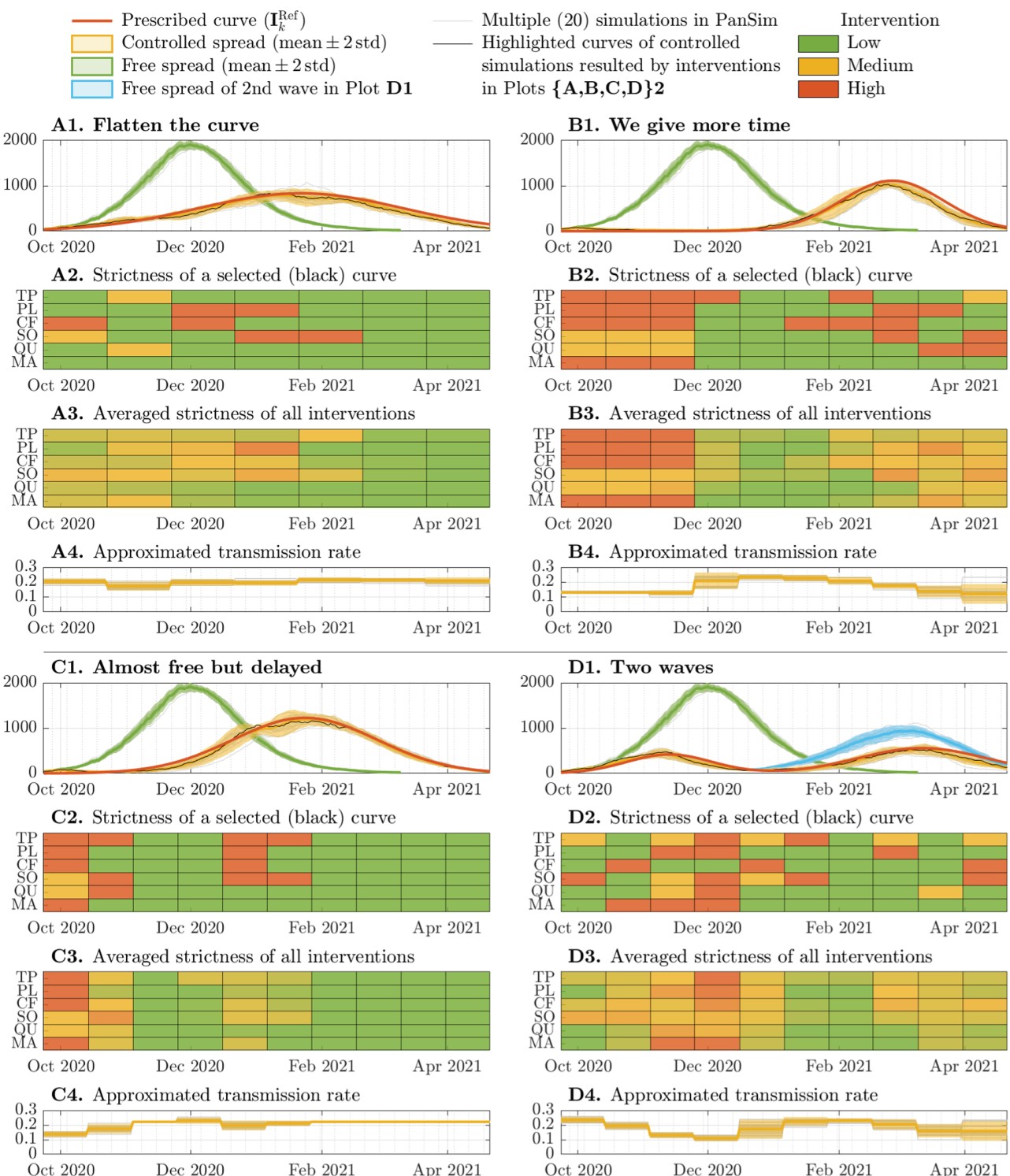

**Fig 2. Free and controlled spread of a well-characterized pathogen with four different control goals determined by a target curve for the number of infected people in the main phase of the disease with symptoms ($I_k$) in a 100k population.** The four cases are distinguished by labels **A**, **B**, **C**, and **D**. The first plots (**A1, B1, C1, D1**) in all cases illustrate the achieved curves obtained through multiple executions compared to the targeted curve and the free spread. In each case, we highlight in black the obtained curve, which is closest to the mean of all executions. This curve was obtained using the interventions presented in the second plots. The third plots illustrate the average strictness of interventions computed through the multiple MPC executions. The fourth plots illustrate the achieved transmission rates. Labels TP, …, MA on the $y$ axes of the second and third plots correspond to the six policy measures explained in Table 1. The mean and standard deviation of each curve were computed from 20 separate executions.

In all studies, we designed a target curve for the numbers of active cases (**I**), which is meant to be tracked using NPIs. However, our framework allows to formulate multi-objective strategies as well. For example, instead of following a target curve, we can minimize both the active cases and the impact of the applied interventions simultaneously. In the simplest case, the impact of the an NPI can be modeled by the resulting transmission rate drop, otherwise, we can define an economic/social impact value for each NPI.

The following four titles labeled by capital letters **A**–**D** denote sub-subsections of the current subsection, each corresponding to a case study.

**A. Flattening the curve.**   First, we would like to delay the peak by 7 weeks with a maximum value of 836 (out of 100,000), namely, we want to "flatten the curve" as presented in **A1** of Fig 2. The results suggest that the targeted curve can be achieved even with monthly intervention planning cycles ($h = 30$). One possible monthly sequence of NPIs is presented in **A2** of Fig 2. Multiple executions show that mild interventions in the first four months are enough to achieve the desired epidemic curve, then, in the decreasing phase of the epidemic, interventions can be gradually released.

**B. We give more time.**   Secondly, the outbreak is intended to be drastically suppressed in the first three months, to give more time to hospitals for scaling up there capacity. As illustrated in **B3** of Fig 2, this can be achieved by a lock-down in the first 9 weeks, after which the interventions can be released for a short period. Then, when the outbreak emerges, the interventions should be slowly relaunched, to achieve the peak at 1114 (out of 100,000) by the end of week 22. This means that the peak is 12 weeks delayed compared to the peak of the free spread. This strategy does not consider the impact on the economy but is focused only on gaining more time to develop the infrastructure in hospitals.

**C. Almost free but delayed.**   When the impact on the economy is also a crucial aspect, one may plan to delay the peak e.g., by 8 weeks but allow the pathogen to spread almost freely. This strategy requires a lock-down in only the first 3 weeks, and mild relaunch after the recognition of the exponential phase (10.–12. weeks) of the outbreak. Otherwise the interventions can be completely released (see **C1**–**C4** in Fig 2).

**D. Two waves.**   Finally, the fourth strategy will allow the population to celebrate a winter holiday without extreme measures, as the epidemic has already been suppressed by previous restrictions. This strategy also a makes it possible to organize public events later by releasing all mobility restrictions in the first half of February (see **D3** in Fig 2). On the other hand, if we split the epidemic into two waves, each of the two peaks will be smaller than in the first case, "Flatten the curve". This can be a good strategy for a limited number of hospital/ICU beds. Admittedly, decision makers need to keep making closures over the whole period, but they are not very strict. If the second wave is completely released, the peak will not be higher than in the third case, "We give more time".

## Scenarios with significant model uncertainty

Next, we analyze how the controller performs in unexpected situations, e.g., when the emerging pathogen is not well characterized, or when a new variant emerges during a pandemic. We analyze three situations. First, we simulate that a new variant appears in the 11th week, which has similar characteristics to the Alpha variant of SARS-CoV-2 [56]. Secondly, a new variant is simulated appearing in the same week but characterized as an Omicron BA.1-like variant [57]. In the third study, we analyze how the controller can cope with an Omicron BA.1-like pathogen appearing in a fully susceptible population. In all three case studies, we consider a prediction ODE model with parameters calibrated to the wild strain of SARS-CoV-2 virus and prescribe a flattened epidemic curve (as illustrated in **A1** (Fig 2). In these studies, we consider

multiple cycle lengths $h = 7, 14, 21, 30$ days, and analyze how they affect the robustness of the controller, when the model is not well-calibrated. In these three studies, we executed PanSim 50 times for each cycle length. The following three titles labeled by capital letters **E–G** denote sub-subsections.

**E. Alpha variant emerges.** *In silico*, we reproduce the events when the Alpha variant appeared in Hungary. In the decreasing phase of an outbreak caused by the wild strain, a more virulent variant called the Alpha appeared with even higher transmission rate. In a free spread, the new variant appearing in week 11 does not seem to be more impactful than the wild mutant. Therefore, we try reusing the interventions in **A2**, which successfully flattened the epidemic curve for the wild mutant (see **A1** in Fig 2). However, these interventions are insufficient to contain the spread for the emergent new variant, and, as it is illustrated in **E1** in Fig 3, the outbreak peaks 1.5 times higher than the peak of the free spread of the wild mutant.

Then, we considered an MPC strategy, in which the $\beta$ values in the LUT are scaled up at the end of every NPI planning cycle if the estimated $\beta$ is 20% higher than it was preliminarily expected for the selected NPI. The threshold of 20% is about two times higher than the standard deviation of the obtained transmission rates for an arbitrary fixed NPI. Therefore, a more than 20% higher transmission rate lies outside of the 95% confidence interval of the expected rate. The results obtained for $h = 7, 14, 21, 30$ days cycle length are summarized in Fig 3 in **E2**, **E3**, **E4**, and **E5**, respectively. In each of these plots, the first vertical line ("Alpha appeared") shows the date when the Alpha variant appeared. With the second vertical line ("$\beta$ higher than expected"), we illustrate the first time when the computed $\beta$ was as least 20% higher then

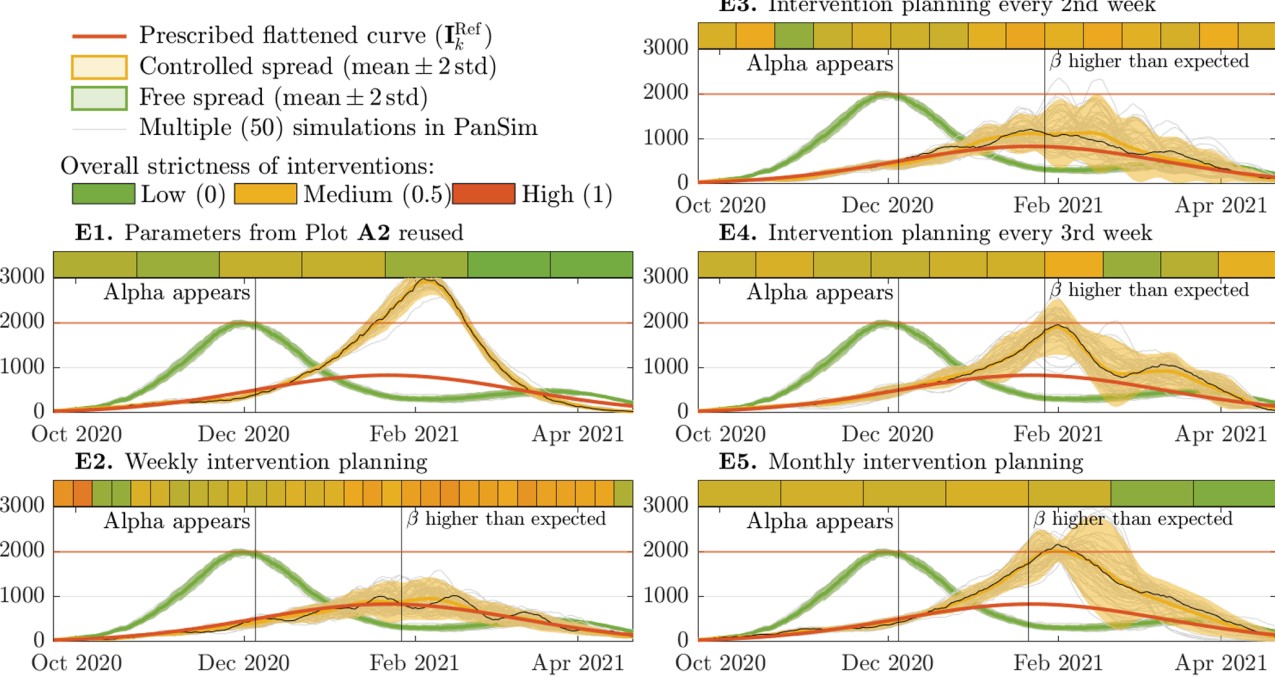

**Fig 3. Free and controlled spread of a pathogen like the wild strain of SARS-CoV-2 and a new variant like the Alpha variant appearing in the 11th week.** The vertical line with the label "$\beta$ higher than expected" illustrates the first cycle, after which the reconstructed transmission rate was at least 20% higher than the average registered in the LUT for the selected and applied NPI. This anomaly suggests that the current pathogen is more infectious than expected.

it was targeted by the applied NPI. In this case study, the controller detects the increase in $\beta$ about 50 days after its appearance, regardless of the length of the planning cycles.

We found that a two weeks long intervention planning cycle is necessary to provide a plausible adaptivity of the controller strategy. When the cycle is $h = 21$ or 30 days long, the controlled spread peaks more than two times higher than it was prescribed by the reference curve. The dates and heights of the peaks as well as the percentage of the population who has been infected in the three unexpected scenarios and control strategies applied are summarized in Table 2.

**F. Omicron variant emerges.**   We assume that the spread is started on day 1 by the wild SARS-CoV-2 strain, then, a new variant like Omicron BA.1 appears in the 11th week. Again, we analyze the free and controlled spread of two variants. The results of the case study are summarized in the five subplots of Fig 4, which are labeled by **F1**–**F5**.

When the spread is not controlled at all, the outbreak peaks in week 10, then, 5 weeks after the appearance of the Omicron variant, rebounds in the middle of the week 16. Up till this time, 60% of the population is infected. One month after the rebound, the second wave peaks at 81% of the altitude of the first peak in week 26. After 210 days, 110% of the population is infected or reinfected.

If we reuse the interventions in **A2** (Fig 2), the spread of the wild strain is suppressed, but the new variant causes a more then twice higher peak compared to the free spread of the wild strain (**F1**). About 93% of the population is infected in this case.

Similarly to the previous case study (**E**), in Fig 4, we illustrated the dates when the algorithm first computed a $\beta$ more than 20% higher than it was targeted. This study reveals that the adaptivity of the control strategy degrades with the increasing length of the intervention planning cycles. Using $h = 7, 14, 21,$ and 30 days long cycles the appearance of a new variant is detected 4, 4, 5, and 7 weeks later, respectively, and the outbreak peaks 1.5, 2, 4 and 5 times higher than as prescribed. We can observe that shorter cycles allows to detect and handle unexpected increase earlier.

**Table 2. Date and height of peaks of outbreaks in the case studies where the emergent pathogen is not well characterized.** The third column called *cumulative infected* (CI) indicates the percentage of population who have been infected or reinfected.

| Mitigation strategy | Peak(s) [no.ind./100k] (week) | | Cumulative infected (CI) in % of population | |
|---|---|---|---|---|
| *E. Alpha variant appears in week 11 (Fig 3)* | | | at rebound | by the end of the period |
| Free spread | 1977 (wk 10) | 481 (wk 26) | two peaks, rebound in wk 20, CI 61% | CI 76% |
| NPIs in **A2** reused | 2917 (wk 20) | | | CI 87% |
| MPC every week | 900 (wk 18) | | | CI 47% |
| MPC every 2nd week | 1114 (wk 17) | | | CI 57% |
| MPC every 3rd week | 1927 (wk 19) | | | CI 70% |
| MPC every month | 2030 (wk 19) | | | CI 79% |
| *F. Omicron BA.1 appears in week 11 (Fig 4)* | | | | |
| Free spread | 1986 (wk 10) | 1615 (wk 20) | two peaks, rebound in wk 16, CI 59% | CI 110% |
| NPIs in **A2** reused | 4044 (wk 17) | | | CI 97% |
| MPC every week | 1240 (wk 17) | | | CI 67% |
| MPC every 2nd week | 1923 (wk 16) | | | CI 82% |
| MPC every 3rd week | 2823 (wk 17) | | | CI 95% |
| MPC every month | 4071 (wk 17) | | | CI 88% |
| *G. Omicron BA.1 in a fully susceptible population (Fig 5)* | | | | |
| Free spread | 7828 (wk 5) | | | CI 97% |
| NPIs in **A2** reused | 5810 (wk 6) | | | CI 97% |
| MPC every week | 1401 (wk 8) | | | CI 88% |
| MPC every 2nd week | 1541 (wk 7) | | | CI 92% |

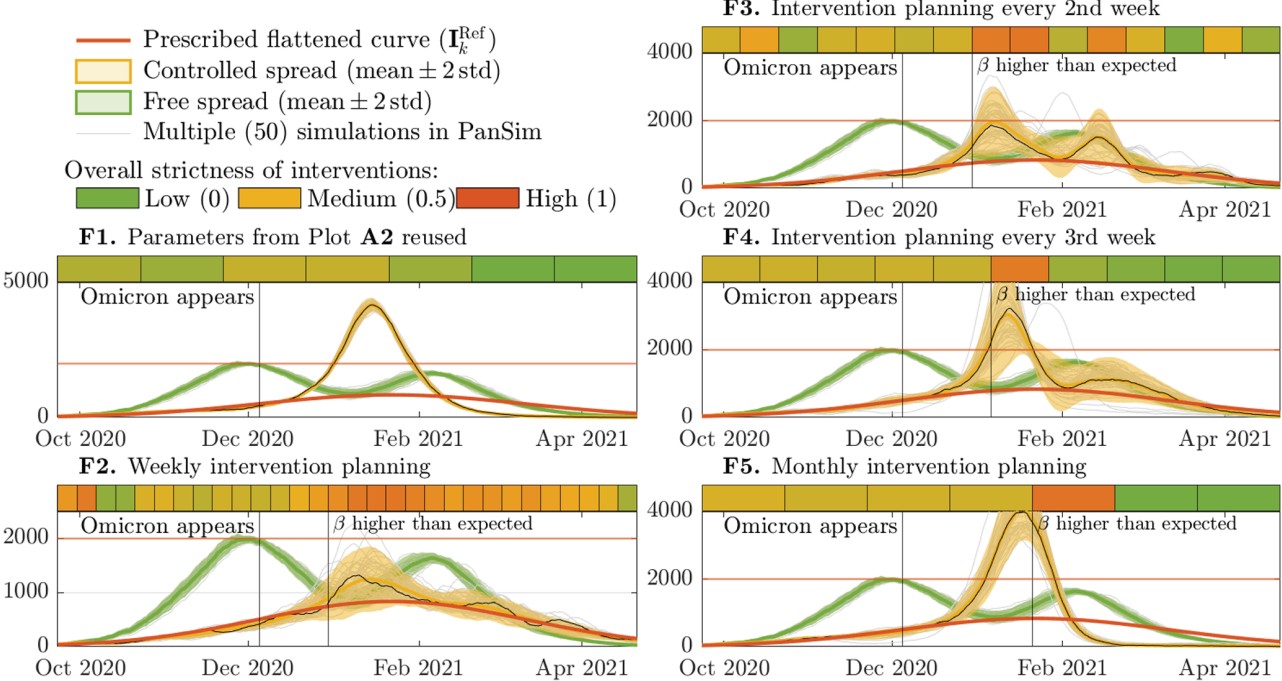

**Fig 4. Free and controlled spread of a pathogen like the wild strain of SARS-CoV-2 and a new variant like the Omicron BA.1 appearing in the 11th week.** The vertical line with the label "$\beta$ higher than expected" illustrates the first cycle, after which the achieved transmission rate was at least 20% higher than it was targeted.

**G. Omicron appears in a fully susceptible population.** In this case, we considered a new pathogen with a transmission rate and disease characteristics similar to the Omicron variant of SARS-CoV-2. Namely, this is the case when it turns out that both the control model (ODE) and the simulation model (ABM) are highly uncertain, as we do not know precisely its transmission rate nor its time ($\tau_\bullet$) or probability ($p_\bullet$) constants. We assumed that this pathogen emerges in a fully susceptible population. When the spread is not controlled, almost the full population is infected in 6 weeks. For this pathogen, the controller is not able to flatten the curve to the desired level even with an $h = 7$ days long intervention planning cycles. In all cases, the higher transmissibility is detected after the first cycle, after which the controller imposes a full closure to achieve the desired targets. Although a full closure has not been able to reduce the spread of the epidemic significantly, an $h = 7$ days cycle length allows to reduce the number of cases from 97% to 88% of the population. These results are illustrated in Fig 5.

## Discussion

The main motivation of this work is to make epidemic intervention-planning more explicit. For this, we address the challenge of integrating advanced model-based control and high-fidelity, realistic agent-based simulation. To handle the various constraints on the state and input variables, we use a model predictive control (MPC) strategy. The controller uses an 8-compartment ODE-based model which is discretized in time. The manipulable input of the model is the spreading rate of the disease which can be influenced between certain limits by appropriate measures. MPC can take into consideration a main practical limitation, namely that the strictness level of measures can only be changed after an interval of several days or even weeks. The cornerstone of the realistic nature of the study is the PanSim simulator [5],

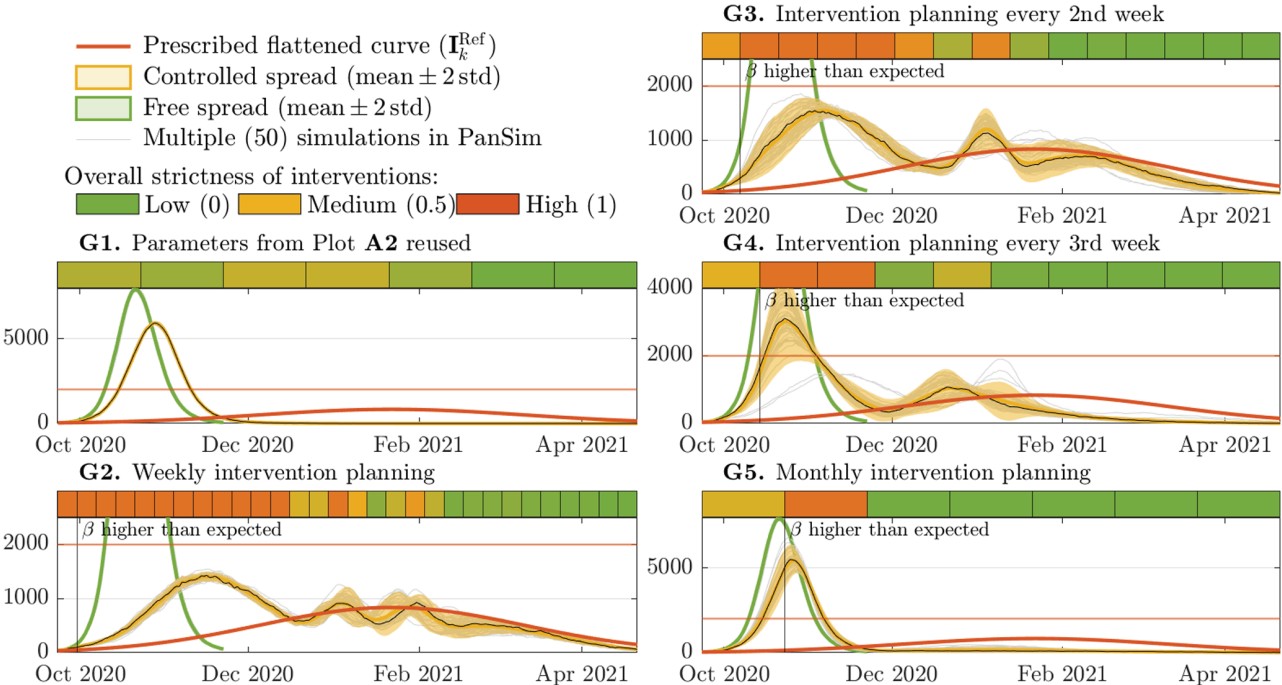

**Fig 5. Free and controlled spread of a pathogen similar to the Omicron BA.1 in a fully susceptible population.**

which is a state of the art agent-based model capable of tracking the everyday behaviour of hundreds of thousands of agents in an urban environment. Several workplaces, institutions, public places and living districts are implemented in PanSim which allows the detailed tracking of infection chains and also the simulation of various measures and quarantine policies. The attributes and epidemic states of the agents can also be set and simulated in detail.

In contrast to the closed-form ABM in [37], PanSim can be controlled by an MPC only through an interface built on three key pillars: calibration, reconstruction, and input translation. The first step involves calibrating the parameters of the control model to align with Pan-Sim by minimizing the squared error between the simulated and observed number of infected individuals in the main phase of the disease (**I**). Next, we use an optimization-based approach to reconstruct the simulator's state evolution in terms of the control model's compartments. Since most of the control model's states are not directly measurable in practice, but are essential for effective control, this reconstruction estimates the missing states based on available data. Additionally, this process enables the computation of the actual daily transmission rate of the pathogen. Finally, the spreading rate prescribed by the controller is translated into specific measures using a lookup table generated from numerous stochastic simulations. This table captures the potentially complex and non-additive effects of applying multiple interventions in parallel. By integrating these three components, the interface effectively closes the loop between mathematical control and agent-based simulation.

Several simulation scenarios are presented to illustrate the capabilities of the proposed methodology. The initial step of the design is the definition of a time-varying reference for the infected people which is to be followed by the controlled system. Together with other constraints on the input and states, this allows the prescription of different goals of various computational complexity such as flattening the epidemic curve or timing the peak of the

epidemic wave at a given level. The effect of significant model uncertainties is also studied in the context of the appearance of new, not precisely modeled virus variants with a higher spreading rate than the initial pathogen. The obtained simulation results underline the importance of early response and that of the selected intervention period. The results clearly show that the tracking performance is good and the variance of the obtained infected curves is acceptably small if the mathematical model used in the controller is precise enough. In such cases, even a large intervention period of one month can be enough for successful operation. However, the possibility of the frequent change of measures becomes important when model uncertainties appear. Otherwise, keeping the infected level close to the prescribed value may become infeasible, and the variance of the output may grow unacceptably large. If the transmission rate is very high (like in the case of the Omicron variant in a susceptible population) then it is not possible to reach the control goals even with the application of the strictest modeled measures (for similar practical observations, see e.g., [58,59]).

In the following, we compare our methodology to the most closely related solutions in the literature. Niemann et al. [43,44] implemented an optimization-based predictive controller using a gradient method, in which both the ABM and its surrogate is used as a prediction model. In contrast, our approach does not include the ABM in the control optimization loop, instead, the ODE-based MPC is computed repeatedly (with a gradually decreasing prediction horizon) considering the "updated" state after each simulation cycle ($h = 7, 14, 21$, or $30$ days). Moreover, we put a major focus on considering realistic measures (Table 1), which are easy to implement, whereas, Niemann et al. considered two measures: the proportion of children not attending school and the proportion of adults working home. These two measures belong to a continuous range $[0,1]$, which makes it straightforward to map the control input to an intervention in the ABM, however, in a real-word situation these continuous-range interventions are not practical as it requires selecting a group of people not attending school or work. Moreover, the compartments in both of our ODE and ABM models are more refined compared to the ABM [29] and its SIR-based surrogates used in [43,44], e.g., infected people are split to 8 compartments in [5] and only 4 compartments in [29] according to the phase and severity of the disease. On the other hand, the method proposed in [43,44], relies on a geo-referenced demographic agent-based simulator (GERDA) [29], which alongside its higher flexibility compared to PanSim has over an order of magnitude lower simulation throughput. First, compared to the hourly schedules in GERDA, PanSim uses a shorter, 10 minutes long time step, which facilitates capturing the short-duration activities and infection dynamics, as potentially risky durations of exposure can be as short as 15-20 minutes [60]. Secondly, a 2000 hours (83 days) long simulation on a mid-range Desktop PC in PanSim with 180k agents and 81k locations takes about 27 seconds. This allows a relatively large number of simulation runs which is clearly useful for uncertainty evaluation. For comparison, the execution time of GERDA for 14k agents and 5k locations is about 30 minutes.

To facilitate traditional (static, linear, and in a quadratic sense optimal) state or output feedback design, the central question of [45] is how to construct and parameterize an ODE-based surrogate model. Our methodology is different in that it bypasses this challenge since we already have an ODE model (SEIR-type) that exhibits sufficiently analogous behavior to our ABM. Instead, we focus on designing a robust and predictive control algorithm for intervention planning. Our approach builds on this by integrating ODE-based control with an ABM, bridging the gap between mathematical tractability and real-world intervention strategies.

The main limitations of the approach are the following. We assume constant time periods between the possible changes of measures although there might be a need for the adaptive change of these intervals depending on the actual state of the epidemic. (We remark that the

applied control design method technically allows such an adaptive solution.) Similarly, a finer resolution of the levels of certain measures might provide more precise reference tracking. Additionally, the currently used cost functions do not take into account the socio-economical impact of the infections in different age groups or that of the suggested interventions. However, these factors can be included in the optimization using, e.g., statistical data [4,61] or dynamical model [62]. Moreover, the recursive feasibility of the MPC approach was not analyzed in this paper, although we have to add that infeasible runs or runs with too high variance are also informative, and suggest that the objectives cannot be met with high probability and/or more precise information is needed to handle the epidemic wave.

In summary, our approach is an important step in making mathematical model based epidemic control more realistic. Firstly, it is shown through various scenarios that an appropriately designed feedback can efficiently reduce the effect of model uncertainties which are inevitable especially in the initial phases of outbreaks. Secondly, the practical problems of given intervention periods of possibly several weeks and quantized input can be handled by the applied MPC methodology. Thirdly, the mapping of the controlled input to specific measures is also solved. Moreover, it is simple to re-parameterize the proposed computation and control design scheme for a different disease or virus variant.

Future work will be focused on the further differentiation and extension of the model, and on the optimization of the combinations of possible interventions based on their economic and societal impact.

## Methods

### State equations

The transition dynamics of the epidemic are characterized by the following discrete-time dynamical equations:

$$\mathbf{S}_{k+1} = \mathbf{S}_k - \beta_k\,\mathbf{S}_k\big(\mathbf{P}_k+\mathbf{I}_k+q_{\mathrm{A}}\mathbf{A}_k+q_{\mathrm{H}}\mathbf{H}_k\big)/\mathbf{N}, \tag{2a}$$

$$\mathbf{L}_{k+1} = \mathbf{L}_k + \beta_k\,\mathbf{S}_k\big(\mathbf{P}_k+\mathbf{I}_k+q_{\mathrm{A}}\mathbf{A}_k+q_{\mathrm{H}}\mathbf{H}_k\big)/\mathbf{N} - \tau_{\mathrm{L}}\,\mathbf{L}_k, \tag{2b}$$

$$\mathbf{P}_{k+1} = \mathbf{P}_k + \tau_{\mathrm{L}}\,\mathbf{L}_k - \tau_{\mathrm{P}}\,\mathbf{P}_k, \tag{2c}$$

$$\mathbf{I}_{k+1} = \mathbf{I}_k + p_{\mathrm{I}}\tau_{\mathrm{P}}\,\mathbf{P}_k - \tau_{\mathrm{I}}\mathbf{I}_k, \tag{2d}$$

$$\mathbf{A}_{k+1} = \mathbf{A}_k + (1-p_{\mathrm{I}})\tau_{\mathrm{P}}\,\mathbf{P}_k - \tau_{\mathrm{A}}\,\mathbf{A}_k, \tag{2e}$$

$$\mathbf{H}_{k+1} = \mathbf{H}_k + p_{\mathrm{H}}\tau_{\mathrm{I}}\,\mathbf{I}_k - \tau_{\mathrm{H}}\,\mathbf{H}_k, \tag{2f}$$

$$\mathbf{D}_{k+1} = \mathbf{D}_k + p_{\mathrm{D}}\tau_{\mathrm{H}}\,\mathbf{H}_k, \tag{2g}$$

$$\mathbf{R}_{k+1} = \mathbf{R}_k + (1-p_{\mathrm{H}})\tau_{\mathrm{I}}\,\mathbf{I}_k + \tau_{\mathrm{A}}\,\mathbf{A}_k + (1-p_{\mathrm{D}})\tau_{\mathrm{H}}\,\mathbf{H}_k. \tag{2h}$$

For simplicity, let the epidemic state and the parameter vector be denoted by

$$x_k = \big(\mathbf{S}_k, \mathbf{L}_k, \mathbf{P}_k, \mathbf{I}_k, \mathbf{A}_k, \mathbf{H}_k, \mathbf{D}_k, \mathbf{R}_k\big)^{\top}, \quad \varrho = \big(\tau_{\mathrm{L}}, \tau_{\mathrm{P}}, \tau_{\mathrm{I}}, \tau_{\mathrm{A}}, \tau_{\mathrm{H}}, q_{\mathrm{A}}, q_{\mathrm{H}}, p_{\mathrm{I}}, p_{\mathrm{H}}, p_{\mathrm{D}}\big)^{\top}. \tag{3}$$

The transmission rate $\beta_k$ of the pathogen can be considered as the control input, which can be manipulated by the non-pharmaceutical interventions. The number of infected people $\mathbf{I}_k$ in the main phase of the disease is considered as an output.

To use the compartment model (2) as a control model for the ABM, we need to face the following challenges:

CL1.  Model parameters in $\varrho$ should be estimated as they do not appear explicitly in PanSim, but through intervals or probability distributions. Moreover, the transitions between

the subcompartments of **I**, **H**, **R** in the ABM makes it even more difficult to infer the transition rates and probabilities of the compartments in the ODE model.

CL2.    The stochastic nature of the simulation makes it hard to approximate the average transmission rate of the pathogen.

In the following two sections, we address these two challenges. First, we address Challenge CL1, and propose a method to fit the ODE model to the ABM.

## Model matching

To align the ODE model with PanSim, we calibrate the model coefficients of (2) such that the solutions best match the simulation data. For this, we first collected simulation data, considering an exhausting number of combinations of interventions in the different phases of the outbreak. A total of 1000 simulations were performed, each lasting 168 days. During each execution, we simulated six different intervention packages, with each package applied for $h = 28$ days. Based on the authors' experience, a 4-week period is adequate to reveal the specific effects of an NPI.

Then, the model calibration is performed such that the results of PanSim for $\mathbf{I}_k$ is considered as the "primary reference". Then, we consider another measure $h(x_k)$ as a reference to estimate one single model constant $\varrho_a$ from the vector of constants $\varrho$. Constant $\varrho_a$ is estimated such that the state vector $x_k$ should satisfy the recursion (2) for a given initial state $x_0$ and with constant parameters $\varrho$ (except for $\varrho_a$), furthermore, $\mathbf{I}_k$ and $h(x_k)$ should match the data obtained from the simulator as much as possible in a squared error sense. The estimation of $\varrho_a$ can be formalized as follows.

**Problem 1** (Parameter estimation)**.** Consider a discrete-time epidemic processes model (2) within the time horizon $k \in [0, T]$. Presume an initial epidemic state $x_0$ at $k = 0$. Let $\varrho_a$ denote one distinguished parameter from the vector of parameters $\varrho$, such that all the parameters from $\varrho$ are fixed except $\varrho_a$. Furthermore, consider the following two sequences:

R1.    $\mathbf{I}_k^{\text{Ref}}$ as a reference for the number of people in the main phase of the disease having symptoms,

R2.    $h_k^{\text{Ref}}$ as a reference for a function $h(x_k)$ of the epidemic state,

where $k = 1, \dots, T$. We are looking for

UV1.    the daily epidemic states in terms of the cardinality of the eight compartments $x_k$, $k = 1, \dots, T$,

UV2.    the daily transmission rates $\beta_k \in [\beta_{\min}, \beta_{\max}]$, where $k = 0, \dots, T-1$,

UV3.    the value of parameter $\varrho_a$,

such that the state equations (2) are satisfied and the solution minimizes the following cost function:

$$J = \underbrace{\sum_{k=1}^{T} w_{\text{I}}^{\text{ref}} \mid f\mathbf{I}_k - \mathbf{I}_k^{\text{Ref}} \mid^2}_{\text{tracking error } (\mathbf{I}^{\text{Off}})} + \underbrace{\sum_{k=1}^{T} w_{\text{h}}^{\text{ref}} \mid h(x_k) - h_k^{\text{Ref}} \mid^2}_{\text{tracking error } (h_k^{\text{Ref}})} + \underbrace{\sum_{k=1}^{T-1} w_{\beta}^{\text{s}} \mid \beta_k - \beta_{k-1} \mid^2}_{\text{smoothly varying transmission rate}}, \qquad (4)$$

where $w_{\text{I}}^{\text{ref}}, w_{\text{h}}^{\text{ref}}, w_{\beta}^{\text{s}}$ are the weight parameters of the multi-objective optimization. *(End of Problem 1)*

The unknown variables of the optimization are declared in UV1, UV2 and UV3 of Problem 1. For the weight parameters we used $w_\text{h}^\text{ref} = w_\text{I}^\text{ref} = 1$, $w_\beta^\text{s} = 10^6$ due to the multiple orders of magnitude difference between $\mathbf{I}$ and the slope of $\beta$.

The advantageous topology of the compartmental model (having no cycles in it), allows us to estimate the model constants one by one, such that changing a parameter $\varrho_b$ in a later step does not change the effect of a parameter $\varrho_a$ in an earlier step significantly.

The parameter calibration relies on the observation that the relative cardinality of every compartments depends on the average residence time $1/\tau_\bullet$ in that compartment. For example, a compartment with a shorter residence time empties faster than another compartment with a longer residence time. Consequently, a higher residence time results in a more populated compartment. On the other hand, the relative cardinality of pairs $(\mathbf{I}, \mathbf{A})$, $(\mathbf{H}, \mathbf{R})$, $(\mathbf{D}, \mathbf{R})$ also depend on the probability constants $p_\text{I}, p_\text{H}, p_\text{D}$.

Preliminarily, we fix the average length $1/\tau_\text{I}$ of the main phase of the disease based on serological analysis. We furthermore assume that $q_\text{A}$ and $q_\text{H}$ are known and fixed. Based on the clinical data we fix the average hospitalization time $1/\tau_\text{H}$.

Using these considerations and the preliminarily fixed constants, the parameters of the ODE model (2) are calibrated in the following steps:

Step1. **Align S, obtain $p_\text{I}$.** The proportion between $\mathbf{S}$ and $\mathbf{I}$ is practically affected only by $\tau_\text{I}$ and $p_\text{I}$. Parameters $\tau_\text{L}$ and $\tau_\text{P}$ affect the phase shift between $\mathbf{S}$ and $\mathbf{I}$ but not their proportion. Therefore, to estimate $\varrho_a = p_\text{I}$, we consider the simulated number of susceptibles $h(x_k) = \mathbf{S}_k$ as a secondary reference and solve the optimization in Problem 1. We note that the time-dependent variable $\beta_k$, which determines the shape of the epidemic curves is also searched alongside the parameter $p_\text{I}$.

Step2. **Align L, obtain $\tau_\text{L}$.** After fixing $p_\text{I}$, we consider $h(x_k) = \mathbf{L}_k$ as a secondary reference to estimate $\varrho_a = \tau_\text{L}$.

Step3. **Align P, obtain $\tau_\text{P}$.** After fixing $p_\text{I}$ and $\tau_\text{L}$, we consider $h(x_k) = \mathbf{P}_k$ as a secondary reference to estimate $\varrho_a = \tau_\text{P}$.

Step4. **Align A, obtain $\tau_\text{A}$.** Constant $p_\text{I}$, which strongly influences the ratio of $\mathbf{A}$ to $\mathbf{I}$, is already calculated and fixed, hence, $\varrho_a = \tau_\text{A}$ is calibrated by fitting $h(x_k) = \mathbf{A}_k$ to the simulation results.

Step5. **Align H, obtain $p_\text{H}$.** The proportion between $\mathbf{I}$ and $\mathbf{H}$ is affected both by $p_\text{H}$ and $\tau_\text{H}$. However, $\tau_\text{H}$ can be well estimated from the clinical data, therefore, $\varrho_a = p_\text{H}$ can be estimated from the simulated results for $h(x_k) = \mathbf{H}_k$.

Step6. **Align D, obtain $p_\text{D}$.** Finally, the simulated number of deceased people $h(x_k) = \mathbf{D}_k$ is used to estimated $\varrho_a = p_\text{D}$.

We considered the following values: $1/\tau_\text{I} = 4$ days, $q_\text{A} = 0.75$, $q_\text{H} = 0.1$, $1/\tau_\text{H} = 12$ days. For $p_\text{I}, \tau_\text{L}, \tau_\text{P}, \tau_\text{A}, p_\text{H}, p_\text{D}$ we reused the values considered in [48] for the wild strain as initial guess. Then, through the parameter calibration, we obtained $p_\text{I} = 0.48$, $1/\tau_\text{L} = 1.5$ days, $1/\tau_\text{P} = 3.1$ days, $1/\tau_\text{A} = 4.1$ days, $p_\text{H} = 0.076$, $p_\text{D} = 0.48$.

## Unknown input and state reconstruction

In order to simulate a real epidemic situation, we assume that the transmission rate and the number of infected individuals not showing symptoms (i.e., those in compartments $\mathbf{L}, \mathbf{P}, \mathbf{A}$) are unknown. However, we presume the knowledge of the exact number of symptomatic patients in the main phase of the disease ($\mathbf{I}$). As we already discussed, this assumption is not

unrealistic, but it facilitates the presentation of the essence of the control approach presented in this paper.

In the following problem formulation, we propose a dynamical model-based data assimilation technique to compute average transmission rates per intervention planning cycle, and simultaneously estimate the daily epidemic state $x_k$.

**Problem 2.** Presume an initial epidemic state $x_0$ at $k = 0$. It is furthermore given a reference sequence $\mathbf{I}_k^{\text{Ref}}$ for the number of people in the main phase of the disease having symptoms, where $k = 1, \dots, T$. Assume that the time horizon $[0, T]$ is split into $p$ intervention planning cycles ($p \ll T$) of length $h$, i.e., $ph = T$. We are looking for

UV1.  the daily epidemic state in terms of the cardinality of the eight compartments $x_k$, $k = 1, \dots, T$,

UV2.  the average transmission rate $\bar{\beta}_j \in [\beta_{\min}, \beta_{\max}]$ in each cycle $j = 1, \dots, p$,

such that the epidemic state equations (2) are satisfied with

$$(\beta_0, \dots, \beta_{T-1}) = (\bar{\beta}_1, \dots, \bar{\beta}_p) \cdot M \tag{5}$$

and the solution minimizes the following cost function:

$$J = \sum_{k=1}^{T} w_k \left\| \mathbf{I}_k - \mathbf{I}_k^{\text{Ref}} \right\|^2, \tag{6}$$

where $(w_k)$ is a sequence of weights. Furthermore, $M \in \mathbb{R}^{p \times T}$ is predefined sparse matrix, in which sum of each column is 1. *(End of Problem 2)*

Note that in Problem 1, the transmission rate was searched as a slowly variable daily series $\beta_k$, where $k = 0, \dots, T-1$. In contrast, in Problem 2, the transmission rate is computed as a piecewise constant staircase function, such that we are searching for the mean transmission rate $\bar{\beta}_j$ for each cycle $j = 1, \dots, p$. E.g., when $h = 7$ days, $\bar{\beta}_j$ denotes the weekly mean transmission rate. This mean value is assumed to be a good estimate for all days in the corresponding cycle, therefore, in Problem 2, we are looking for the same $\beta_k = \bar{\beta}_j$ rate for every day $k = h(j-1), \dots, hj-1$ of each cycle $j$. This consideration is formalized in (5), where the $k$th column of matrix $M$ determines, which value of $\bar{\beta}_j$, $j = 1, \dots, p$ has to be used on day $k$. When the interventions are presumed to take effect promptly, matrix $M$ is defined as follows:

$$M = \begin{pmatrix} 1 & \dots & 1 & 0 & \dots & 0 & & 0 & \dots & 0 \\ 0 & \dots & 0 & 1 & \dots & 1 & & 0 & \dots & 0 \\ & \dots & & & \dots & & \dots & & \dots & \\ 0 & \dots & 0 & 0 & \dots & 0 & & 1 & \dots & 1 \end{pmatrix} \in \mathbb{R}^{p \times T}. \tag{7}$$

However, this matrix allows us to model a gradual transition in $\beta$ between two interventions, e.g., a 2-day long transition can be described by the following matrix:

$$M = \begin{pmatrix} 1 & \dots & 1 & 0.7 & 0.3 & 0 & \dots & 0 & & 0 & 0 & 0 & \dots & 0 \\ 0 & \dots & 0 & 0.3 & 0.7 & 1 & \dots & 1 & & 0 & 0 & 0 & \dots & 0 \\ & \dots & & & & & \dots & & \dots & & & & \dots & \\ 0 & \dots & 0 & 0 & 0 & 0 & \dots & 0 & & 0.3 & 0.7 & 1 & \dots & 1 \end{pmatrix} \in \mathbb{R}^{p \times T}. \tag{8}$$

The uncertainty analysis of the reconstruction is presented in Sect 4 of S1 Text.

## Construct the lookup table (LUT)

To construct a LUT, we identify how the distinct NPIs affect the average transmission rate in an $h$ = 28 days long cycle. Through an unknown input state reconstruction (Problem 2), we approximate the average transmission rates resulted by the several NPIs simulated through the 1000 random executions. In the LUT, an average transmission rate and its standard deviation are assigned to each NPI. During the average computations, we omit those simulated days where less than 0.5% of the population is infected. The constructed LUT can be considered an input translation between the agent-based and compartmental ODE models or as a mapping between the relative transmission rate and the NPIs.

The standard deviations of the computed transmission rates obtained for the different NPIs are between 5% and 25%. This means that the effects of some NPIs are more uncertain and result in larger variations in the $\beta$ than other NPIs whose effects can be predicted more accurately. However, the standard deviations in $\beta$ computed for the 216 NPIs has a median of 10%. Therefore, we chose 20% as the confidence bound, in the sense that if the (retrospectively) calculated $\beta$ is more than 20% higher than the average value in the LUT, we attribute this to changes in pathogen properties. Otherwise, the anomaly in $\beta$ can be credited to the stochastic nature of the process. The uncertainty of the entries in the lookup table is quantified in the supplementary material (S1 Table) and present in Sect 3 of S1 Text.

## Control loop

The control algorithm has three major steps and one optional:

   I.  Intervention planning for the future cycles.
  II.  Apply the selected NPI for the next cycle.
 III.  Unknown input reconstruction and state observation for the past cycles.
  IV.  Update the mapping $\beta \mapsto$ NPI.

The steps of the control loop are illustrated in Fig 1.

**Step I. Intervention planning.** Let $c \in \{0, h, 2h, \dots, (p-1)h\}$ denote the current time at the end of an intervention planning cycle. Let $q \in \{1, \dots, p\}$ denote the next future cycle, $c = (q-1)h$. We presume the knowledge of the current state $x_c$. Then, we solve Problem 2 from the initial state $x_c$ within the time frame $k = c+1, \dots, T$, such that $\mathbf{I}_k^{\text{Ref}}$ constitutes the target curve for compartment $\mathbf{I}$ to be achieved. The program computes a sequence of transmission rates $\bar{\beta}_{\text{d},j}$ for each upcoming cycles of the epidemics $j = q, \dots, p$. The first computed value for the transmission rates ($\bar{\beta}_{\text{d},q}$) is used as the targeted value for the next cycle. Utilizing the statistical model (i.e., the LUT), we select multiple NPIs that is expected to achieve a transmission rate close to the target through different strategies. This offers an additional degree of freedom in the control design, allowing for flexibility in choosing the most appropriate intervention. To ensure a minimal deviation between NPIs from cycle-to-cycle, we first identify NPIs whose (e.g.) the 67% confidence interval (mean $\pm$ std) for $\beta$ contains the target rate $\bar{\beta}_{\text{d},j}$.

Note that Problem 2 was formulated as an unknown input state reconstruction, nevertheless, it can be reused as an MPC program with the declared modifications.

**Step II. Apply interventions.** The selected NPI is applied in the following cycle on days $k = c+1, \dots c+h$. When the control strategy is executed as an experiment, the NPI is simulated on the agent-based model using PanSim and the values for compartment $\mathbf{I}$ are collected as a measurement.

We remark that the MPC has computed transmission rates for the whole control horizon (for multiple future cycles), yet only the first control action is applied in the first future cycle. The remaining values $\bar{\beta}_{d,q+1}, \dots, \bar{\beta}_{d,p}$ will be recalculated in the upcoming intervention planning steps.

**Step III. Reconstruction.** Using the measured values for **I** as reference $\mathbf{I}_k^{\text{Ref}}$, we solve the program in Problem 2 in time frame $k = 1, \dots, c + h$ to compute the new state $x_{c+h}$ and the actual transmission rate $\bar{\beta}_q$ achieved in the recent cycle $q$.

In real pandemic event, we collect the number of patients in hospitals, the number of daily hospital admissions, and the weekly viral loads measured in the wastewater treatment plants. These data allow a detailed reconstruction of the spread according to our enhance reconstruction method presented in [48]. This method computes not only $\mathbf{I}_k$, $k = 1, \dots, c + h$ but also $x_{c+h}$, and $\bar{\beta}_q$. Considering the computed $\mathbf{I}_k$ as a measurement, we propose to execute a second reconstruction but now according to Problem (2). The second reconstruction may improve the accuracy of the predictions in the MPC for the following reasons. The epidemic model used in [48] is slightly different from (2) not including the waning immunity nor the viral load. Moreover, we found that a precise and detailed epidemic model gives bad predictions if the present state is not well-estimated. Similarly, we demonstrated in [48] that a bad epidemic model can produce good predictions if the present state is estimated according to the same model.

**Step IV. Update the LUT.** If the transmission rate $\bar{\beta}_q$ computed retrospectively is 20% higher than its expectation $\bar{\beta}_{d,q}$ (according to its actual mean value in the LUT), the transmission rate value in each entry of the LUT is increased proportionally by the one third of the measured rate increment. Namely, the transmission rate values in the LUT are multiplied by $1 + (\bar{\beta}_q/\bar{\beta}_{d,q} - 1)/K$, where $K = 3$. (This corresponds to an exponential forgetting with factor $\lambda = \frac{K-1}{K} = \frac{2}{3}$.)

## Overview and summary

It is noteworthy that the control input computation, intervention selection, reconstruction, and LUT update are executed only once in a cycle. But the time frame, in which the consecutive steps are executed is different. To clarify the time schedule of the algorithm in a single intervention planning cycle, we present a short summary of the time frames in the following equation:

$$
\begin{array}{c}
\text{I. Intervention planning} \Rightarrow \bar{\beta}_{d,q} \\[2pt]
\overbrace{\qquad\qquad\qquad} \\
\text{cycle of length } h \qquad \text{present time } (x_c) \\
\overbrace{\qquad} \qquad\qquad \downarrow \\
\left| \; j=1 \; \right| \; \dots \; \left| \; j=q{-}1 \; \right| \; \blacksquare \; \left| \; j=q \; \right| \; \dots \; \left| \; j=p \; \right| \\
\underbrace{\qquad\qquad} \\
\text{II. Apply the selected NPI in the next cycle} \\[4pt]
\underbrace{\qquad\qquad\qquad\qquad} \qquad \uparrow \\
\text{III. } x_{c+h} \text{ and } \bar{\beta}_q \text{ reconstruction} \quad \Big| \\
\qquad\qquad\qquad\qquad \text{IV. Update LUT}
\end{array} \tag{9}
$$

The vertical lines in (9) separate the $p$ intervention plannings cycles, the thick vertical line illustrates the current time, at which we plan a control action (I) and apply during cycle $q$ (II). When cycle $q$ comes to its end, we reconstruct the evolution of the spread (III), and scale the transmission rate values in the LUT if necessary (IV).

The time schedule suggests that the overall time frame of an epidemic control mission is fixed to $T = ph = 210$ days, such that the horizon of the MPC is decreasing from cycle to cycle since the end date of the horizon is not advanced with the time. We often call this implementation a *decreasing horizon* MPC. However, this algorithm can be easily modified to a so-called *receding horizon* MPC, if the horizon is advanced by $h$ days from cycle to cycle.

Also note that the reconstruction is implemented in an increasing horizon fashion as the starting date of the reconstruction is not advanced in time. This is reasonable until an outbreak is evolving, however, when the number of infected people decreases below a given threshold, it may be useful to reset the starting date of the reconstruction to that date.

Throughout the paper, we described how we connected a robust controller to the complex microsimulation model, PanSim. The interface between the MPC and the simulator was implemented at three levels: parameter calibration, state reconstruction, and control action translation. First, we calibrated a compartmental ODE model to match the simulator's dynamics by minimizing the squared output error, as detailed in Problem 1. Second, at each cycle, we reconstructed the simulator's entire history and current state using measurements from a single observable quantity. Third, the computed control actions were translated into specific combinations of non-pharmaceutical interventions (NPIs) using a lookup table, updated with each cycle.

## Supporting information

**S1 Text. Alongside the hardware information of our framework, the supplement includes multiple figures illustrating the computed mean and variances in the lookup table, and the uncertainty of the reconstruction, and finally, it presents a control execution of Scenario A ("Flattening the curve") in more pictures.**
(PDF)

**S1 Table. This supplement provides the numerical values of the statistical information in the lookup table, which maps a transmission rate to a combination of interventions.**
(XLSX)

## Author contributions

**Conceptualization:** Péter Polcz, István Z. Reguly, Kálmán Tornai, János Juhász, Sándor Pongor, Attila Csikász-Nagy, Gábor Szederkényi.

**Formal analysis:** Péter Polcz.

**Funding acquisition:** Attila Csikász-Nagy, Gábor Szederkényi.

**Investigation:** Péter Polcz.

**Methodology:** Péter Polcz.

**Project administration:** Attila Csikász-Nagy.

**Resources:** István Z. Reguly.

**Software:** Péter Polcz, István Z. Reguly.

**Supervision:** Gábor Szederkényi.

**Validation:** Sándor Pongor, Attila Csikász-Nagy, Gábor Szederkényi.

**Visualization:** Péter Polcz.

**Writing – original draft:** Péter Polcz, István Z. Reguly, Gábor Szederkényi.

**Writing – review & editing:** István Z. Reguly, János Juhász, Sándor Pongor, Attila Csikász-Nagy, Gábor Szederkényi.

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
