## [Decision Letter · Decision Letter 0]

16 Dec 2024

PCOMPBIOL-D-24-01801

Smart epidemic control: A hybrid model blending ODEs and agent-based simulations for optimal, real-world intervention planning

PLOS Computational Biology

Dear Dr. Polcz,

Thank you for submitting your manuscript to PLOS Computational Biology. After careful consideration, we feel that it has merit but does not fully meet PLOS Computational Biology's publication criteria as it currently stands. Therefore, we invite you to submit a revised version of the manuscript that addresses the points raised during the review process.

Please submit your revised manuscript within 60 days Feb 15 2025 11:59PM. If you will need more time than this to complete your revisions, please reply to this message or contact the journal office at ploscompbiol@plos.org. Please include the following items when submitting your revised manuscript:

We look forward to receiving your revised manuscript.

Kind regards,

Alejandro Fernández Villaverde, Ph.D.

Guest Editor

PLOS Computational Biology

Hannah Clapham

Section Editor

PLOS Computational Biology

**Additional Editor Comments :**

All three reviewers made suggestions that entail substantial changes in the manuscript. I think that it is feasible to address these suggestions, which would lead to an improved paper. That said, some of these suggestions would require considerable additional work that may be excessive. Specifically, Reviewer #1 suggests modifying the model to another pathogen and comparing the results to COVID, and Reviewer #2 suggests comparing the proposed method with another control strategy. While those studies would indeed be interesting, and the authors are welcome to pursue them if they wish to, I don't think that they should be essential requirements for the acceptance of the paper.

**Journal Requirements:**

At this stage, the following Authors/Authors require contributions: Péter Polcz, István Z. Reguly, Kálmán Tornai, János Juhász, Sándor Pongor, Attila Csikász-Nagy, and Gábor Szederkényi. Please ensure that the full contributions of each author are acknowledged in the "Add/Edit/Remove Authors" section of our submission form.

4) Please amend your detailed Financial Disclosure statement. This is published with the article. It must therefore be completed in full sentences and contain the exact wording you wish to be published.

2) State what role the funders took in the study. If the funders had no role in your study, please state: "The funders had no role in study design, data collection and analysis, decision to publish, or preparation of the manuscript.".

If you did not receive any funding for this study, please simply state: u201cThe authors received no specific funding for this work.

5) Please ensure that the funders and grant numbers match between the Financial Disclosure field and the Funding Information tab in your submission form. Note that the funders must be provided in the same order in both places as well. Currently, the order of the grants is different in both places.

Please indicate by return email the full and correct funding information for your study and confirm the order in which funding contributions should appear. Please be sure to indicate whether the funders played any role in the study design, data collection and analysis, decision to publish, or preparation of the manuscript.

**Reviewers' comments:**

Reviewer's Responses to Questions

Reviewer #1: The authors prevent a hybrid ODE and agent-based model for determining what combination of interventions would be required to achieve a desired epidemic trajectory, and suggest that their approach could be adapted to other diseases.

Major comments:

First, the model seems to be "fit for purpose" as demonstrated. However, I kept wondering what the advantage was of such a complex approach. Most of the interventions described can be fairly simply described in terms of impact on either the number of contacts or (nearly equivalently) the transmission rate (beta). If you need to reduce the number of infections by 30% to meet your target, can't you just choose (rather than simulate) the set of interventions that will get you very close to that? Relatedly, I would be curious to know how well just the ODE model performs on these tasks without including PanSim.

Second, the authors mention "it is straightforward to modify the parameters and fit the ABM to a different pathogen" (p. 7). This seems unlikely to me for a couple reasons. First, this is almost never as easy as it seems! Second, the code seems to be written in C++ and Matlab, which are both suitable choices for high-performance simulations, but neither of which are that familiar to the majority of epidemiologists (though the authors are thanked for making their code public, and it seems well-written). The authors don't discuss how long the model takes to run or what compute resources are required, but I imagine it is significant if a GPU implementation is warranted. Finally, if the model is "straightforward to modify", the authors are encouraged to modify it to another pathogen (e.g. mpox, or another non-respiratory pathogen of relevance to their work) and compare-and-contrast the findings with COVID.

Minor comments:

p. 5: 10 min is a very small timestep -- how much do results differ if the timestep were an hour or even a day?

p. 5: 10% forecast accuracy 3 weeks in advance isn't bad. However, given that hospitalizations usually lag infections, often much simpler measures (such as test positivity rate) can also get good performance predicting hospitalizations.

p. 6: "Secondly, our study focuses on managing the epidemic over a period of about six months, assuming that the initial virus strain does not mutate during this time. Therefore, it is reasonable to assume that the effects of the non-pharmaceutical interventions are time-invariant." I think time-invariance is a reasonable assumption, but I don't think this is why. The impact of, say, school closures, changes the number of contact events that occur. The effect of reducing contacts should be time-invariant regardless of "mutations" (and, indeed, new variants certainly can arise within 6 months).

p. 7: "Then, the LUT is filled in with the averages of the resulting transmission rates." Why not fit a Gaussian process emulator or similar? Wouldn't this be more efficient, more accurate, or both?

p. 10: "One possible monthly sequence of NPIs is presented" -- sorry if I missed it, but what happens when the same outcome can be achieved in multiple different ways (e.g., school closures or curfew, but without needing both)?

Reviewer #2: The authors present a study on model predictive control (MPC) applied to the simulated phases of the COVID-19 pandemic in Szeged. They employ a highly detailed COVID-19 agent-based model of Szeged, previously published in PLOS Computational Biology. This model is used to both estimate unknown variables and evaluate the impact of a prescribed set of non-pharmacological interventions (NPIs).

The primary limitation of the study is the lack of real-world data to validate the control algorithm. However, this is justifiable given that real-world testing would require waiting for a future pandemic, which is impractical.

Overall, the manuscript is well-written, and the approach is both interesting and innovative. Nevertheless, I believe the study would be further strengthened by addressing the following points:

-Quantify the accuracy of the state reconstruction method.

-Quantify the accuracy of the lookup table method.

-Comparison with alternative approaches: If feasible, comparing the proposed method with another control strategy could demonstrate its relative strengths or limitations.

Minor Suggestions:

-Line 129: Remove the extra period.

-Figures: Improve figure captions to make them more detailed and enable independent interpretation without referring to the -main text.

-Line 360: In silico should be italicized.

-Line 522: Rephrase "In the authors’ experience" to make it more formal and objective.

-Line 588: The phrase "Somewhat, we want to simulate a real epidemic event" could be revised.

-Consider reducing the use of abbreviations, as the text contains many that might hinder readability for a broader audience.

Reviewer #3: Summary of Work

This paper addresses the issue of how to optimally choose interventions to control an epidemic. Specifically, they consider a detailed agent-based model (ABM) which contains extensive details of the spread of the pathogen as well as the effect of potential interventions. The full space of policy choices is considered too complex and computationally expensive to explore using the agent-based model. To solve this problem, they pair the complex model with a simple ODE model where the effect of interventions can be extensively analysed using model predictive control (MPC) strategies.

Major Considerations

The exact contribution of this paper was unclear with the message often getting lost amongst the details. The papers needs to be dramatically shortened (i.e. 10 pages as opposed to 26), with the novel contribution and methodology being clearly articulated, and unnecessary details removed to Supplementary Materials. For example, whilst the paper is about how to choose policy, it is not a policy paper discussing what policies should be used under which circumstances, rather it is a methodology paper. Therefore a detailed examination of scenario A, contrasting the results with other methodologies and developing a detailed understanding of why the results were as they were, would be far more impactful than a list of scenarios A-G.

The idea of having a simple surrogate model which is paired with an expensive detailed model is not novel. There is an extensive literature on using emulators in this way (often as part of a calibration method). The methodology of this paper needs to be compared to these established methods, with its advantages/disadvantages examined.

The paper talks about using an ODE model, but from my understanding of the equations in the methodology it is a discrete time compartmental model not an ODE model. If this is the case, then it should be described as such throughout.

One aspect of the epidemic which was key in 2021 was it’s age stratified nature (e.g. see positivity rates in the UK data which could be different by a factor of over 10 between the age groups). With vaccination rates being vastly different between age groups, along with certain NPIs affecting different age groups (i.e. school closures), I don’t see how a non-age stratified model can be an appropriate emulator of the full system. Explaining how the proposed methodology handles these situations is important to evaluate the validity of the approach.

From my understanding of the methodology, the parameters (rho) of the simplified model are estimated once for the whole simulation. Given that the ABM is a much more rich model, my expectation would be that they would vary over time depending upon changes in the ABM states (e.g. vaccination rates, closures of specific types of institutions), which I expect this would give a better fit of the short-term dynamics within each policy period. The paper needs to examine the effect of only having a single set of model parameters.

**Have the authors made all data and (if applicable) computational code underlying the findings in their manuscript fully available?**

Reviewer #1: Yes

Reviewer #2: Yes

Reviewer #3: **No: **Whilst a link to a Github repository for the underlying ABM was provided, I could not find the code related to the work presented in this paper.

PLOS authors have the option to publish the peer review history of their article (what does this mean?). If published, this will include your full peer review and any attached files.

Reviewer #1: No

Reviewer #2: No

Reviewer #3: No

**Figure resubmission:**
---

## [Decision Letter · Decision Letter 1]

7 Apr 2025

Dear Dr. Polcz,

We are pleased to inform you that your manuscript 'Smart epidemic control: A hybrid model blending ODEs and agent-based simulations for optimal, real-world intervention planning' has been provisionally accepted for publication in PLOS Computational Biology.

Before your manuscript can be formally accepted you will need to complete some formatting changes, which you will receive in a follow up email. A member of our team will be in touch with a set of requests. During completion of that task, **please take the opportunity to make the following minor modification** in the ABSTRACT: change "Ordinary differential equation (ODE) models" to "Ordinary difference equation (ODE) models". This is to make the definition of the acronym given in the Abstract consistent to the one given in the Introduction, which is used throughout the paper. This modification was suggested by one of the reviewers; since it only entails changing one word, I have considered it unnecessary to ask formally for even a "minor revision". You may ignore the other concerns communicated by the reviewer at this point. 

Best regards,

Alejandro Fernández Villaverde, Ph.D.

Academic Editor

PLOS Computational Biology

Hannah Clapham

Section Editor

PLOS Computational Biology

Reviewer's Responses to Questions

**Comments to the Authors:**

Reviewer #1: The authors are thanked for their extremely thorough response. I have no further comments.

Reviewer #2: The revisions have effectively addressed my comments.

Reviewer #3: For the response to comments addressed by the authors I use their numbering from the response letter.

Comment 13: My concerns around this comment still stand. This paper has potential as a methods paper, but not as a policy paper (e.g. comments 17/18 demonstrate why the analysis of some of the interventions is invalid).

Comment 14: The authors have added a literature review of similar papers/methods in the field to the Introduction and have discussed how they differ in methodology in the the Discussion.

Comment 15: Using ODE as an acronym for ordinary difference equations is non-standard, but if consistently used is okay. Please can you update the Abstract to reflect this (currently the acronym is defined differently in the Abstract and Introduction).

Comment 16: Thank you for clarifying that the age-dependent details are captured by the underling ABM. Whilst it is possible that a reasonable emulator might be possible without some age-structure, this would have an impact on the static assumptions of parameters (see response to Comment 17).

Comment 17: I disagree with the assumption and justification that most of this parameters would be time-independent in the emulator. One of the key features of COVID was its age-structure, especially when considering disease progression e.g. the probability of a symptomatic infection (pI) and probability of hospitalisation (pH) changes by orders of magnitude depending on whether it is a young child or an elderly person. Given that the emulator has no age-structure, the true epidemic dynamics can only be captured if these are time-dependent, reflecting on periods when the epidemic is concentrated in children versus spread evenly throughout all age groups. This is especially true as some of the policy interventions analysed explicitly target only some age groups (e.g. School Closures).

Comment 18: Thank you for making the code available on Github.

**Have the authors made all data and (if applicable) computational code underlying the findings in their manuscript fully available?**

Reviewer #1: Yes

Reviewer #2: Yes

Reviewer #3: Yes

PLOS authors have the option to publish the peer review history of their article (what does this mean?). If published, this will include your full peer review and any attached files.

Reviewer #1: **Yes: **Cliff Kerr

Reviewer #2: **Yes: **David Henriques

Reviewer #3: No

---

## [Editor Report · Acceptance letter]

PCOMPBIOL-D-24-01801R1

Smart epidemic control: A hybrid model blending ODEs and agent-based simulations for optimal, real-world intervention planning

Dear Dr Polcz,

I am pleased to inform you that your manuscript has been formally accepted for publication in PLOS Computational Biology. Your manuscript is now with our production department and you will be notified of the publication date in due course.

With kind regards,

Zsofia Freund
